# The Paradox of Chivalric Madness: Ariosto's and Cervantes's Madness Representations' Impact on Disability Representation

Nicholas L. Johnson

Department of Comparative Literature, College of the Liberal Arts, Pennsylvania State University, State College, PA 16801, USA; nlj5145@psu.edu

**Abstract:** This study investigates the connection between madness and critiques of the chivalric romance genre in two late Renaissance works, Ludovico Ariosto's *Orlando Furioso* and Miguel de Cervantes's *Don Quijote de la Mancha*. The satire of chivalric romance in these works of fiction caution against nascent modes of thinking in imperial societies for the implementation of chivalric ideas to inspire and promote imperial conquests in Latin America through juxtaposition with the Muslim and Moorish conquest in the Maghreb and through metaphorical island governance. In order to make such critiques, these novels implement the madness of their parodic knights to disguise their critiques. This practice establishes a precedent which later literature can employ to make sociocultural critique covertly, to the detriment of disability representations as literary devices or metaphors.

**Keywords:** early modern Europe; disability studies; mad studies; madness; chivalry; coloniality; satire

> In the domain of literary and philosophic expression, the experience of madness in the fifteenth century generally takes the form of moral satire. —Michel Foucault, *Madness and Civilization: A History of Insanity in the Age of Reason*

> But if the essential world-views implicit in the concept of art. . .are in every respect revealed by art, then art has got rid of this content which on every occasion was determinate for a particular people, a particular age, and the true need to resume it again is awakened only with the need to turn *against* the content that was alone valid hitherto. . .in Italy and Spain, when the Middle Ages were closing, Ariosto and Cervantes began to turn against chivalry. —G.W.F. Hegel, *Aesthetics: Lectures on Fine Art*

## 1. Introduction

Ludovico Ariosto's *Orlando furioso* and Miguel de Cervantes Saavedra's *Don Quijote de la Mancha* use mad knight figures to symbolize concern with outdated chivalric romances' resurgence during the enactment of imperialism on other continents as a means of resolving the Hapsburg empire's political and economic crises and, in the later work, the epistemological development of rationalism and empiricism in the Baroque period. Madness representation functions in both plots to sublimate satirical commentary on chivalric romances as outdated social structures, setting a precedent for madness's perception in perpetuity due to these narratives' renown. The issue taken with this implementation is its subjugation of madness as a literary device in order to advance plot or to contribute to overarching themes. Meanwhile, disability studies scholar Michael Bérubé asserts that disability can be represented justly only if the disabled character's disability does not contribute to the plot. In other words, the character's disability can remain a part of their identity as long as it is not a means to a literary end ([Bérubé 2005](#)). This does not occur with these Early Modern narratives; meanings assigned to disability result from metonymic devices which conceal sociocultural critiques in a time of censorship. Connoting negative social value with disability rather than the intended paradigmatic conflict risks distracting readers from said conflict, instilling the idea that disability instead is the source of societal troubles.

In what ways do *Orlando Ffurioso* and *Don Quijote de la Mancha* employ madness? What might critical disability inquiry, such as Elizabeth Bearden's *Monstruous Kinds*, offer to the discourse surrounding these Early Modern representations? How does the use of madness reflect or impact Early Modern epistemology? What do their respective impacts on literary and societal norms engender for disability representation and perception in the Renaissance and Baroque periods? I contend that in these two narratives, madness embodies a paradox helpful to the elaboration of literary and social progress when seen as a manifestation of Bearden's definition of *sprezzatura* in Early Modern European society; disability is both a social liminality and a necessity centralized in European discourse.

The study of chivalric madness begins with an examination of disability, satire, and Early Modern epistemology to highlight the function disability has for shielding satire and political critique of the burgeoning imperial system, which in turn contextualizes Elizabeth Bearden's likening of *sprezzatura* to disability masking in early modern court life and how it structures disability representations in the texts under investigation. Next, I turn to Ariosto's influences and intertextual world. The study then proceeds to link chivalric romance and madness: how etiology is constructed to justify Orlando's madness in a religious–moral model of disability, the impact of chivalric madness upon the greater social order, and the resolution of Orlando's madness and his subsequent surprise at the result of his pursuit of chivalric social expectations as a tension with Bearden's *sprezzatura* as disability masking. Then, this study outlines intercultural connections between Ariosto and Cervantes to explain why Cervantes takes on the trope of the mad knight to pursue similar social criticism to that of Ariosto. Cervantean chivalric madness also allows social critiques to evade censors, which engages with the question chivalric romances' relevance in late sixteenth- and early seventeenth-century Spain to the imperial project. Then, the study critically examines Don Quijote's madness etiology, and finally proceeds through a showcase of his chivalric madness's effects on both himself and society. This leads to a discussion of Cervantes's metafictional commentary as another form of criticism embedded in, and bolstered by, its chivalric madness. Through this, literary madness as a vehicle to critique early imperialist structures, agents, and philosophies becomes a literary device.

## 2. Analysis

### 2.1. Madness, Satire, and Early Modern Europe

Early Modern European society centered on royal courts, and understandings of disability, satire, and the Enlightenment concern the political maneuverings required to successfully endure the societal changes of this period. First, I will explore the interrelationships between imperialism and empiricism as they appear in the epistemological and literary record. Next, the use of satire to evade censorship and connect with the audience will connect these Enlightenment ideas to the use of *sprezzatura* as outlined by Elizabeth Bearden to adjust one's social perception. From here, it will be clear that the mad knights of Arioso and Cervantes are a form of literary *sprezzatura* to call into question the developing imperial logics of their time.

The empirical focus on the measurability of the world coming into being as Ariosto wrote his poetry commends itself to the mastery over the natural world. This mastery over nature inherent to empirical thought—and by extension, the presumed superiority over those who have not mastered nature to an identical extent—allowed the European to distinguish the human body from nature. Desiderius Erasmus typified humanity through the capacity to reason as a distinction from the natural world: "This is why he is called a rational being, and this is what sets him apart from animals. And what is the most harmful influence upon man? Surely it is ignorance" (Erasmus 1978). The conflation of reason and unnaturalness parallels the connection between madness and animality, as drawn from Medieval precedent and Christian teleology (Thiher 1999, pp. 46–47). Furthermore, Elizabeth Bearden's use of Gleeson's geographies of desire in the early modern context relies on the construction of monstrosity as disability due to a relationship between monstrosity and nature before this period:

The degree to which people are depicted as monstruous in the premodern tradition relies heavily on how the natural climate they occupy is viewed and affects them, and how they civilize or are made savage by the spaces in which they dwell. Gleeson's observations, then, regarding the accumulative effects of historico-geographical frames on disability across different times and spaces enliven the spatial link between monstrosity and disability. (Bearden 2019, pp. 23–25)

Bearden remarks that the "monstruous narrative" of the Renaissance as "texts inhibiting narratives with delayed or lost time, with nonlinear chronologies, or with shifting, inaccessible, or wayward topologies are typical of crip time and space", which accurately describe both texts in this study, suggest that the natural–unnatural dualism becomes a significant concern of European early modern epistemology (ibid., pp. 27–28).

From here, the compulsory able-bodiedness of the Cartesian mind–body split, temporally coincident with Cervantes, and its ties to the preeminence of Christianity, place the European human self at the center with nature at its periphery (McRuer 2006; Rummel and MacPhail 2021; Bristow 2023). In the imperial context, this self-centeredness lent itself to the generation of eurocentrism that disregarded alternative ways of interacting and experiencing the world that permitted dehumanization of populations in Africa, Asia, and the Americas for not mastering nature as the Europeans had done. Interactions with Moors in both texts under investigation here are expressions of concerns over the imperial that goes hand in hand with the empirical becoming prominent in Early Modern epistemology.

With all these developing ideas, control of dissemination through censorship became the norm for imperializing European kingdoms, and to navigate this increasingly proscriptive, empirical imperialism, many had to invoke *sprezzatura*, a way of carrying oneself that obfuscated how their bodies did not reach the standards of the European body, which was placed at the top of the hierarchy justifying colonization and imperialism. *Sprezzatura* as Bearden constructs it resembles disability masking, or the downplaying of what would render someone aberrant or deviant from the social norm, and so in the literature, *sprezzatura* also functions as a means by which characters rid themselves of the mad, social critique persona into the normative persona when their madness is eventually cured. Explained etymologically, *sprezzatura* concerns "making something difficult seem easy. . .as a technique by which courtiers can compensate for a lack of natural *grazia*, or grace" (Bearden 2019, p. 56). In a way, *sprezzatura* operates prosthetically to the extent that allows one to appear to have mastered their own nature, but its major usage in discounting the significance of disability to one's life points to *sprezzatura*'s reclamation of the very idea of one's nature. Bearden's example of Guidobaldo Montefeltro's disregarding the significance associated with his impairments reinforces a questioning of conventions of the body in the Renaissance (ibid., p. 59). In a similar vein, the implementation of madness and satire in Ariosto and Cervantes's texts enables a questioning of empiricist and imperial paradigms without censorship.

To contend with several centuries of chivalric history, the two satires under investigation were calling into question the usefulness of chivalric romance and its value system to the day's nobility rather than chivalry itself. *Don Quijote*, with its contemporary *Carajico-media*, particularly appeals to a literate and powerful audience by questioning their own practices: "a veiled denunciation of the nobility for its failure to live up to the standards of chivalry" (Domínguez 2015, p. 139). Chivalry then works as a sign in itself for noblemen and the cultural practices of the Early Modern elite, who were beginning to profit from imperial regimes and question the validity of monarchs. Indeed, satire of this period demonstrated "the intellectual preparation and power of the aristocracy in the first part of the sixteenth century, and. . .the ways in which authors employ a system of signs they hold in common to satirize each other" as a means to reflect on imperial values when the Early Modern period introduced several novelties in goods and ideas (ibid., p. 228).

Satire held a prominent position in Early Modern Europe to consider imperial novelties before integrating them into society. Particularly, medical satire and health discourses surrounding the validity of empiricism compared to Galenism contributed to discussions

of the value of imperialism. In her study of guaiac as a medicinal import from Spain's early colonies, Ivana Bičak posits that "satire incorporates current affairs into its literary fabric, taking its inspiration from and reacting to the protagonists, actions and events in the empirical world" and that "the special significance of satiric productions that engage with medical themes lies in the inventive combinations of the literary reality and the empirical reality" (Bičak 2023, pp. 1–2). Satire enables an author to consider epistemology alongside contemporary events, which suggests that Ariosto and Cervantes employed satire, drawing on madness as a medical theme, to think about chivalry in its Early Modern imperial context. This is due to satire's "mediating the empirical world. Moreover, satire is firmly intertwined with the art of medicine and healing since it frequently uses medical rhetoric and imagery to diagnose the society's ills" (ibid., 9). Through her analysis of Cristóbal de Castillejo's *En alabança del palo de las Indias, estando el la cura dél*, Bičak demonstrates the interconnections between medical themes, imperial novelties, and contemporary contentious discourse that works to address Early Modern epistemologies such as Galenism and empiricism without fear of censorship or significant backlash from detractors. As a result, Ariosto and Cervantes's satires play with *sprezzatura*, or lack thereof, in chivalric figures turned mad by the dissonance between chivalric ideals and the Early Modern imperial context.

### 2.2. Ariosto's Chivalric Romance Deleterious to Madness's Perception

How does *Orlando furioso* depict madness, and what does it risk for elaborated madness perceptions in the Early Modern period? The poem employs a thematic relationship between madness and chivalry to use satirical precedents and extend the satire of feudal society. Ariosto's poem contends that chivalry is an outdated principle for social hierarchization and uses madness as a vehicle to demonstrate chivalry's anachronism. Chivalric and mad signifiers become inextricable from one another due to the effectiveness of Ariosto's blending of these tropes. It prepares literary madness representation for future (ab)use and perpetuation as a negative and liminal subject-position in European culture to call into question other societal problematics.

*Orlando furioso* builds off two preceding narratives. The first is *La chanson de Roland*, a unifying text for French nationalism, depicting Roland (Orlando in Italian) leading a small group of knights to defend France from Moorish invaders until Charlemagne's army can arrive (Gautier 1885). However, the second text, *Orlando innamorato*, is not a chivalric romance but a satire of the chivalric tale in which Orlando falls in love (Boiardo 2018). The change in representation from chivalric romance to satirical poems indicates shifting attitudes toward chivalry, and *Orlando furioso* furthers this satirical perception through unrequited love.

Boiardo and the French legend are not the only influences on Ariosto's text. In fact, scholarship traces madness representation back to Dante's *La Divina Commedia*. Noting that *Orlando furioso* is not the starting point of literary madness supports the conviction that madness in the poem has "a rather markedly metaliterary character" relating the poem hermeneutically to Dante (Bartoli 2017, my translation). In other words, any study of Ariosto must be aware of Dante's madness representation and perhaps even earlier writers to arrive at the metaliterary substance of the poem sublimated into Orlando's representation.

*Orlando furioso* opens in an intertextual allusion to Virgil's opening of *The Aeneid* by declaring its themes to the reader: "of ladies, cavaliers, of love and war,/of courtesies and of brave deeds I sing" (Ariosto 1975, I.1.117). Stating its primary concern as ladies and cavaliers over the abstract themes connoted with such identities indicates the poem's interest in the characters whose lives the sociocultural system of chivalry dictates. In addition, the ordering of ladies before cavaliers then asserts a gendered if not feminist critique of chivalric society to be forthcoming. Furthermore, no mention of madness is made in these opening lines, which marks Orlando's madness as a narrative prosthetic, juxtaposing other themes to articulate a critique (Snyder and Mitchell 2014).

Angelica's drinking from the fountain instigates the narrative as a paradigm shift from chivalric romance's conventions. When Rinaldo drinks from the fountain that augments

love and Angelica from the other, that diminishes love, Angelica makes the choice to leave Rinaldo because she does not love him (Ariosto 1975, I.78.136). Angelica then flees, and several knights, including Orlando, pursue her. The plot structure here suggests not only that the knights perceive a woman and her affections as possessions to claim but also that Angelica's divergent affections are a form of gendered deviance from chivalric romance norms. This is reaffirmed in Angelica's choosing Medoro, a Moorish man, to love rather than a Christian knight, because it suggests Angelica's choice, and therefore her subjectivity, breaks down the value-laden chivalric dichotomies of Christian–Muslim and knight–conquered when the potion suppresses her love while choosing a romantic partner, which, given Angelica's origins from either Cathay or India, creates a resistive version of conquered others.

In the Early Modern period, Spanish and Italian depictions of North Africa as *Africa Devicta* referred to their conquest by the Hapsburg empire with the potential of being converted to Catholicism. Always a woman, Africa in Spain and Italy wept due not only to her defeat in battle but also to the prospect of her people no longer following Islam (Baskins and Llopis 2022, pp. 226–27). The combination of conquest and conversion "aimed to restore the 'true' religion and the unity of faith not only in the Mediterranean but around the globe" (ibid., p. 240). As a result, Angelica's choice of Medoro goes against European imperial constructions not only of a mourning feminine alterity, because Angelica's choice indicates a non-conversion to Christianity of Angelica as a metaphorical extension of the gendered land, but also of conquered peoples, in an unconverted Moorish man being worthy of Angelica's affection. The consideration that Angelica's choice of Medoro is a more rational one because love is not involved after her drinking the potion in her deliberation destabilizes the construction of the European Christian as superior to the North African and Asian Muslim. Revealing the fragility of chivalric sociocultural conventions when presented with gendered resistance and religious alterity to domination, *Orlando furioso* gives way to the onset of madness in the amorous knight when not all is as it should be for romance in the burgeoning empire.

Orlando's madness representation emphasizes unrequited love and subsequent frustration as its impetus. The following passage is the onset of Orlando's madness: "He neither sleeps nor eats; though three days pass,/...His grief so swells, his sorrows so amass/that madness clouds him, in which long he erred/...The mail and armor from his back he tore" (Ariosto 1975, XXIII.132.725). His madness comes not from acting on anything related to chivalry directly but from his expectation of love, a sign that knights no longer demonstrate high medieval sociocultural status, and instead is "precisely his nobility of soul that renders him an ideal and impractical character...as if he were in exile, a prisoner of ways and habits foreign to his new everyday life" (Ciccarelli 2017, p. 669, my translation). This display of madness indicates chivalry's anachronism in Ariosto's sixteenth century ideas. Therefore, the scene depicts a knight shedding his armor, the societal marker of his status, and abandoning the chivalric code to adjust himself to "a world firmly upside down...where 'goodness' is only fruit of convenience. If chivalric society is therefore a satirical mirror of actual truth, that reality does not correspond to ideals is no less than obvious" (ibid., p. 669). Chivalry's convenience is made plain when Angelica asks her courters what they gain from adhering to chivalry after its obsolescence:

> O Count Orlando, O Circassian,
>
> Of what avail your prowess and your fame?
>
> What price your honor, known to every man?
>
> What good of all your long devotion came?
>
> Show me one single favor, if you can,
>
> What recompense, . . .
>
> For suffering for her sake undergone? (Ariosto 1975, XIX.31.591)

Angelica's question interrogates not chivalry directly but its byproducts, prowess and fame, which, according to her question, are no longer acquired in the same manner. Orlando's madness, then, is an adjustment to circumstances that demonstrate his mindset has become outdated, associating madness with an anachronism in the move toward the empirical and imperial paradigms.

Orlando's subsequent actions designate a volatile and inhuman quality to madness's alterity. Specifically, he continues to pursue Angelica without adhering to the chivalric code, lusting after her when he sees her, "just as a dog after its prey would race" (Ariosto 1975, vol. 2, XXIX.61.195). The metonymy of madness conveying anachronistic social values takes on the animalistic connotation, which does not appear in the madness of *La Divina Commedia* but is a central component of the understanding of madness in this period. In fact, comparatist analysis suggests the basis for madness's metonymy seeks to play on the words involved in its emplotment through hyperbole. In other words, Orlando does not simply become "furious" at Angelica's choosing Medoro, but he becomes "mad" as the hyperbolic extremity of the Italian word, *furioso*. Due to the ironic nature of this hyperbole, this play on words engages the poem's satire to obfuscate Ariosto's criticism of chivalry under the enveloping determinant of madness (Bartoli 2017, p. 80). The resulting conceptualization incorporates the animalistic qualities of Orlando's chase, a common feature of madness portrayals in the Early Modern period as previously established, with the anachronism of chivalry in this literary madness.

The blending of animality as distinguished from humanity and the hyperbole of anger as madness mixes further in the following canto, when Ariosto's narrator steps back from the narrative to apologize to female readers for Orlando's misogynistic remarks. The narrator justifies the comments as being due to madness, crossing the signifiers so as not to indicate overtly that knights had come to expect love and loyalty from women within the societal structure and substantiate a condemnation of the former set of societal values:

> But I am like a sick and ailing man
>
> Who, after suffering for many years. . .
>
> He yields to rage, and curses all he can.
>
> The pain subsides: his anger disappears.
>
> Aghast he lies repentant on his bed,
>
> But what was said cannot be now unsaid. (Ariosto 1975, XXX.2.199)

Instead, Ariosto attributes the negative elements of Orlando's characterization to madness. However, when understood through *sprezzatura*, this passage redirects us to the failure to embody *grazia* through a yielding to his natural desires, which are not animalistic at all but socially constructed expectations of love. In addition, Ariosto writes reasons for Orlando's madness, which invoke the moral–religious model of disability:

> But your Orlando for his gifts has made
>
> To his Creator but a poor return.
>
> The more it was his duty to lend aid,
>
> The more the Faithful have been left forlorn.
>
> His blinding passion for a pagan maid
>
> This Christian knight of judgment has so shorn
>
> . . .
>
> And God for this has caused him to run mad. (ibid., XXXIV.64–5.328)

In these octaves explaining Orlando's madness, once again a comparison to animals is drawn, but this metaphor is now embedded in religious narratives and connected to a critique of Orlando's performance as a knight to demonstrate his lack of *grazia*. The

first section presents Orlando's actions as detrimental and contrary to Christian values, through which *sprezzatura* criticizes lending aid to the faithful as a practice. In Ariosto's context of early imperial actions, pursuing the "pagan" maid Angelica resembles engaging in cultural exchange with colonized subjects rather than the Christian civilizing mission in the sixteenth century. The ramification is a "punishment" from God for his misbehavior, invoking the religious subsidiary of the moral model of disability. This model suggests that all disabilities are the punitive result of decisions and actions taken which transgress moral codes, established by God, in order to rectify the sociocultural order (Models of Disability: Types and Definitions 2020). This perception of disability, common to Ariosto's time, stigmatizes disability as a negative consequence of one's actions, but it also serves subtly to question the priorities of imperial Europe. The semiotics surrounding madness in *Orlando furioso* have been jumbled to deceive censorship and appease the patrons who commissioned the poem, but people designated as "mad" become the undesirable social outcasts as a result.

A journey to the moon ensues to find Orlando's wits. This journey to the moon reaffirms the metaliterary influence of Dante on Ariosto, demonstrating Ariosto's elaboration on Dantean themes. Placing Orlando's wits on the moon alongside intangible, irretrievable items such as "tatters of fame…prayers to God…the moments lost in empty games of chance,/fruitless projects none could ever realize, [and] the fruitless idleness of ignorance", intimates that without the magical realism a narrative can possess, people cannot restore their sanity once it is lost to their desires for love, power, and fame, while also evidencing Foucault's connection between madness and idleness, since idleness as well as madness appear on the moon (Ariosto 1975, XXXIV.74–5.330; Foucault 1988, p. 46). In any event, the restoration follows the progression of the narrator's apology to women but involves more than that, beginning with a ritualistic cleansing of his body, by washing seven times to remove the dirt (Ariosto 1975, XXXIX.56.447). The removal of dirt furthers the distinction between humanity and nature through a cleansing of dirt as a metaphor for nature to establish this process as one that ought to "purify" Orlando of his madness, suggesting that madness cannot possibly endure for someone of such social status, in that *sprezzatura* enables "disability" passing at court. After the purification takes place, Orlando regains his wits in the next octave, "His intellect returned to its pristine/Lucidity as brilliant as before" (ibid., XXXIX 56–7.447). The madness resolves as suddenly as it began, suggesting that adherence to the new colonial paradigm instead of questioning it restores one's *grazia*. Additionally, Orlando must process what he has done—a clear indication that the ideal chivalric figure Orlando represents in his "regained" self, the self presented in *La chanson de Roland*, would never let his lust take priority over doing good; in other words, the true knight would never question the imperial regime reviving chivalric value, and instead madness as disability is to blame (ibid., XXXIX.58.447).

In summary, madness becomes negatively connotated rather than the anachronistic thinking chivalric romance represents in the poem, via the religious–moral model of disability. The satire also conceals its social critiques underneath the catch-all metonymy of madness-as-disability, because the religious–moral model perceives disability solely as an avoidable and punitive life condition. Miguel de Cervantes invokes the same protection by representing yet another mad knight in his satire, eternalizing and perpetuating negative connotations with madness which arrived there only through a mixing of signifiers in an attempt not to anger patrons.

### 2.3. Connecting Ariosto to Cervantes

How does the literary influence of Ariosto on Cervantes elevate literary madness as a commentary on Early Modern society? There is a great deal of scholarship regarding the intertextual relationship between Ariosto and Cervantes, even though a century occurred between their publications. One critic goes so far as to support this paper's hyperbolic claims about *Orlando furioso* while also asserting that Spanish comedy cited this phenomenon frequently:

His madness no longer belongs to him, it is common to all jealous people who push their passion to its climax. The fury of the paladin is evoked so frequently in the Spanish *comedia* that I refuse to give a summary of its allusions here. (Chevalier 1966, p. 418, my translation)

A common trope contemporaneous with Cervantes in Spanish comedy is a knight driven to madness by unreciprocated love, ensuring Ariosto's impact on madness representations going forward. More aligned with the topic here, however, is Thomas R. Hart's introduction to his comparative text, *Cervantes and Ariosto*, where he instantiates Cervantes's awareness of Ariosto and chivalric romance's symbolic atemporality, which then gets insinuated into the madness representations of the genre's satires as the epigraph of this paper states (Hart 1989, p. 4; Hegel 1975, pp. 591–92). Ariosto and Cervantes both go about emplotting chivalric madness as a conduit for criticism and satire. Madness, within a century of Ariosto, becomes a signifier of satire rather than the signified of chivalry and other outdated ideas, shifting Ariosto's initial implementation of madness.

However, Hart's comparatist analysis reveals another layer to Ariosto's and Cervantes's texts, pointing to metanarrative allusions. Hart draws on Harry Levin, whose interpretation of Don Quijote's actions to emulate and redeem chivalric romance does not view the madness–sanity dichotomy as one devoted to social critique but indicates a divide between art and nature—in other words, fiction and reality (Levin 1957, pp. 79–96). Hart's position on this outlook accepts Levin's view as plausible, while contextualizing this art–reality dichotomy as part of the discourse on supporting outdated social mores because "the pattern of art is in both cases formed by the conventions of chivalric romance, just as nature is the pattern of life in an early modern Europe still breaking away from its medieval past" (Hart 1989, p. 39). As commentary on the anachronism, the metanarrative qualities are tied to the metaphor of chivalric madness and its effectiveness at questioning empirical imperialism and its engagement of chivalry despite censorship in the sixteenth and seventeenth centuries.

### 2.4. Cervantes Borrows from Ariosto for Social and Literary Criticism

How does madness in *Don Quijote de la Mancha* build off Ariosto's precedent and create a perception of disability in the seventeenth century? Miguel de Cervantes Saavedra's text relies heavily on the precedent of Ariosto's poem to construct its protagonist and sublimate imperialist critiques. In addition, it critiques the liaison made at the turn of the seventeenth century between chivalry and Spanish imperialism as a way to realize Erasmian ideals (ibid., pp. 41–43). Cervantes offers an imagination of how attempting global peace through chivalric and religious empire would manifest, sublimating in the protagonist's madness an undesirability to this approach to progress. This commentary draws on Ariosto's disability representation to argue this and further entrench disability into a central role in literature as a sign for undesirable social systems.

While chivalry had not been part of Spanish society for centuries before the writing of *Don Quijote*, Cervantes is writing in a time when chivalric romances and ideals are being revived to inspire conquistadors in colonial, imperial pursuits. In discussion of Bernal Díaz del Castillo's *la historia verdadera de la conquista de Nuevo España*, Cervantes's contemporary paradigm demonstrates the logical leaps from chivalric codes, chivalric romances, and the needs of imperial Spain: "in the 16th century, sent to the Indies were a large quantity of Spanish-language books. . .fiction literature and, more particularly, chivalric romances were the object of exportation from Europe destined for America" (Grunberg 1979, p. 113, my translation). The conquistadors were inundated with chivalric romances in order to promote "patriotic sentiments and. . .chivalric virtues" (ibid., p. 114). The chivalric romances brought with them ideas of nationalism that spurred conquistadors to imperial conquests. Cervantes wrote *Don Quijote* with a response to this promotion of chivalric values being brought into the seventeenth century by showing someone, inspired by chivalric romances to excess, being unable to function in contemporary society under the moniker of madness.

This study has proposed that avoiding censorship added a utilitarianism to madness in Ariosto's and Cervantes's texts, so it is prudent to discuss at the beginning of the analysis of *Don Quijote de la Mancha* the royal documentation in the text's initial pages. These paratextual pages explain the royal support for publishing and selling the novel. Acquiring such permission served a dual purpose: first, it was an early version of legal copyright, protecting Cervantes's text from being printed by anyone else.[1] More importantly, however, it was a means for the monarchy to control the content published and disseminated in the kingdom from being dissident or seditious against sociopolitical structures (Cervantes 2005, I.3–5). Criticizing knights as symbols of conquistadors and the new imperial hierarchy evaded censorship by using madness as an explanation for any negative connotations attributed to the chivalric, imperial figure. With this context, Erich Auerbach's contention that "the theme of the mad country gentleman who undertakes to revive knight errantry gave Cervantes an opportunity to present the world as play" supports the idea that madness protects the sociocultural critique of chivalric values' anachronism but gives way to slippery, coterminous significations (Auerbach 1989, p. 15).

As the analysis of *Orlando furioso* began with a discussion of the literary etiology of Orlando's madness, so it does with *Don Quijote*. In contrast with an onset based on a pseudo-feminist and a certainly anti-chivalric presentation of social norms and mores, Cervantes's text begins in a world that has advanced past chivalry for all intents and purposes, such that the only remnants of it remain in chivalric romances. The hidalgo, Alonso Quijano, becomes Don Quijote by reading these chivalric romances and finding in them a lifestyle preferable to the one he leads as a figure of low feudal status:

> And so, by sleeping little and reading much, his brain dried up in such a manner that he came to lose his judgment...in effect, his judgment already killed, he came to have the strangest thought that ever a madman in the world had,...make himself a knight errant and go all over the world...putting himself in occasions and dangers after which, overcoming them, he would be given eternal name and fame. (Cervantes 2005, I.I.29–31, my translation)

The first phrasing of Quijote's condition here is "to lose his judgment", which becomes "his judgment already killed", but develops the connotation of madness only when he has "the strangest thought" to become a knight himself to earn renown. The desire to be a knight, both to serve his kingdom and earn him glory, designates madness, suggesting that the normate society must have abandoned this paradigm to the past.[2] In contrast with Ariosto's link between love and madness for the knight, insanity connects here to yearning for renown, a plausible motivation considering Alonso Quijano's status as an hidalgo, a position without a great deal of influence in a sociopolitical system in flux. He seeks to gain a higher status, like the knights in his novels, but Quijano misunderstands that the very system in which he would gain status does not complement the imperialism that has emerged in the Baroque period. As a result, his *grazia* stands out wherever he goes far beyond the point that Quijote can employ *sprezzatura* to pass as nondisabled.

The discrepancy between the chivalric and enlightenment social systems appears as early as his first adventure as a knight errant. Don Quijote emulates the behavior of the knights in his books at an inn, imagined as a castle, and speaks to prostitutes as though they were noble ladies. However, his antiquated, formal Spanish is incomprehensible to the prostitutes, and the innkeeper mockingly paraphrases verses of chivalric romances to communicate with the mad knight (ibid., I.II.37–9). Don Quijote starts to leave because "what bothered him most was not being seen armed as a knight, for seeming that he could not legitimately go on any adventure without receiving the order of knighthood", as though the scene in its entirety were not mocking his attempt to be a knight (ibid., I.II.41). The innkeeper and prostitutes ridicule his effort, yet he believes he has not done enough to embody the chivalric code and so retreats. Such is the discrepancy in this passage: aspiring to the highest moral version of chivalry and simultaneously ridiculing that notion of chivalry in contemporary Spain.

We are aware of the reality of Quijote's surroundings only because the narrative voice is third person omniscient and explains the contemporary reality Quijote ignores, suggesting that Quijote's perception is mad to the multiple fictional narrators as well. Cervantes plays on Ariosto's device of an intervening narrator, proposing Cide Hamete Benengeli as a Moorish historian of Don Quijote's narrative, which parallels the Muslim presence in Ariosto's text as a rapprochement of the European self and the colonial other (ibid., I.IX.86). However, to establish Benengeli as the narrator, an unidentified first person also narrates. As a result, both narrators have the opportunity to include their opinions into the text, creating a heteroglossia, and any perspectives in the diegesis can be attributed to them both, such as Quijote's mad imaginings and the incongruity with reality. In this case, madness's presumed synonymity with anachronism emerges from the contemporary worldview by way of an emphasis on the narrators' contemporary perspectives.

Another aspect of *Don Quijote de la Mancha* that employs perspective to make a critique is its intertextuality, especially with chivalric romances. The greatest example of this phenomenon is the barber and the priest's discussion of Don Quijote's collection of books. The priest assesses the collection, at first wanting to burn *Amadís de Gaula*, a canonical chivalric romance, calling it "'the first of the chivalric romances printed in Spain, and all the others have taken their principle and origin from this one; and so, it seems to me that, like a dogmatist of a bad sect, we must without any excuse condemn it to fire'" (ibid., I.VI.61). In this passage where the social commentary is applied to common literature of the time, the allegiance constructed between chivalric romance and the chivalry–madness metonymy shifts to incorporate anything incomprehensible into rejected content from the Spanish public's point of view, as personified by the priest and the barber, so as to indicate how ideas of the past, such as chivalric codes, and of the near present (Ariosto's satire of chivalric romance as anachronistic) were being received. Intertextuality in *Don Quijote* plays with perspective so as to demonstrate further the extent to which chivalric romance as a genre had become anachronistic in seventeenth century Spain. Since the obsolescence of chivalry had not fully been accepted due to the expansion of the Spanish empire at the hands of chivalric romance–inspired conquistadors, Cervantes further satirizes by juxtaposing chivalric ideas with grotesque imagery of contemporaneity.

Cervantes's work also plays with the sacred and grotesque literary tropes identified in Bakhtin's *Rabelais and His World*, using bodily injury to emphasize the incongruity of chivalry with contemporary reality (Bakhtin 1984). Bénédicite Torres marks Sancho Panza as the carnivalesque or grotesque body, juxtaposing Sancho's having brought food on the adventure with Quijote's adherence to the "'honor of errant knights to not eat for a month, and if they eat, it's something that they have at hand'" to create a farcical perception of how knights attend to their bodily needs, and Torres asserts this dynamic "between the carnivalesque squire and the Lenten knight, they cover innumerable nuances throughout the work" (Cervantes 2005, I.X.94; Torres 2002, pp. 234–35, my translation). Torres's dichotomy between knight and squire relies on a divide in social class created along the sacred–profane dichotomy. The mad knight's Lenten attitude of righteous, religious fasting compares with the grotesque of Sancho's admitted ignorance: "Since I don't know how to read nor write. . .I don't know nor have I fallen into the rules of the chivalric profession" (Cervantes 2005, I.X.95). Sancho's ignorance serves to progress his characterization's grotesquery, as it is a body "open" to change through learning, whereas Don Quijote's more "closed" mind, already filled with chivalric romances and ideals, must change through praxis rather than discourse throughout the narrative.

When Quijote's closed body reflects on its pain, it is framed within the noble, chivalric ideals that do not cohere to the other characters' experiences of pain throughout the text. For Quijote, "[the] purpose of this suffering is to create and sustain a larger order", which refers to the perpetuation of chivalry through conquest and empire (Andrade 2019, p. 81). For example, the injury that results from the episode with the merchants who demand to see Dulcinea del Toboso before calling her the most beautiful woman in the world gives Quijote an attempt to protect Dulcinea's honor: "'But you will pay for the great blasphemy what

you have said against such beauty as that of my lady.'" However, Don Quijote's suffering comes as a result of his adherence to chivalric ideals, because his armor is too heavy to lift off the ground. His claim that he is stuck on the ground "'for no fault of my own fault, but for that of my horse'" reinforces the idea that the signifiers of chivalric romance in Quijote's reality are to blame for the situation in which he finds himself (Cervantes 2005, I.IV.54). Seen in this way, his aims inspired by chivalric romance subvert the outcome and so result in physical suffering, to complement the cognitive dissonance of upholding outdated ideas in the contemporary world. Meanwhile, Andrade's analysis of physical pain in the text suggests that "as soon as any of the characters (other than Don Quijote) experience physical pain, they exit their roles as soon as possible... their reactions to pain suggest that they are not so different from everyday men and women" (Andrade 2019, p. 88). Effectively, other characters establish an opposition to Don Quijote's pain tolerance, a manifestation of *sprezzatura*, as a means of differentiating their contemporaneity, their *grazia*, from Don Quijote's desire for a chivalric romance of the Spanish empire.

However, Sancho Panza's gradual adoption of Quijote's civics-oriented chivalric relationship to, and therefore a "mad" perception of, the world over the course of both parts of the text evokes the important colonial discourse established under the pretense of madness, which culminates in the governance of Barataria. Nemser's situating *Don Quijote* and other Golden Age texts "as a form of literary *arbitrismo*" on the moral legitimacy of colonialism in the seventeenth century has drawn out the colonial discourse into which Cervantes enters his text, but it has yet to consider the ramifications of the madness that enable such discourse in the narrative (Nemser 2010, p. 3). For example, Sancho's reason for leaving the insular Barataria, only for its island-ness to be destabilized for the reader in the interaction with Sancho's friend, Ricote, in the chapter after Sancho has left Barataria, refers not only to Sancho's aversion to pain and the dangers associated with governing but also to the fact that he has been misled into believing in this governance in the first place by a lack of sleep that aligns with the onset of Quijote's madness at the beginning of the narrative (ibid., p. 5; Cervantes II.LIV). Sancho's sane aversion to pain, his *sprezzatura*, after the invasion threat tempers his onset of madness from being kept awake at the beginning of his governance, but pain tolerance does not keep him from believing in the ideals of his and Don Quijote's mad chivalry enough and espousing a nobler version of colonial governance, which "cannot be seen as separate from governance at home" (Nemser 2010, p. 21). A paradox emerges here in that the chivalric romance and its values are viewed as anachronistic, while the madness that encourages chivalry in the narrative simultaneously advocates for an avant-garde governing style for the entire Spanish empire in its critique.

Having discussed at length the connection between madness and chivalry in *Don Quijote*, the study now turns to the ending, during which madness gives way to sanity, as it becomes apparent to him that the structures he seeks through chivalry are unattainable. To begin, Don Quijote's defeat at the hands of "el Caballero de la Blanca Luna", the stakes of which were the honor and unrefuted beauty of Dulcinea, makes Don Quijote ask for death instead of conceding:

> —Dulcinea del Toboso is the most beautiful women in the world and I the most wretched knight in the land, and it is not Good that my skinniness defrauds this truth. Thrust, knight, your lance and take my life, then my honor will have left me.

> —This I will not do, for certain– said the knight of the White moon–: live, live in your entirety the fame of lady Dulcinea del Toboso's beauty...(Cervantes 2005, II.LXIV.1047)

Quijote's defeat signals a break in his consistency with regard to pain and suffering. He asks for death here, because his aspirations of realizing a contemporary chivalric romance have been thwarted and because no solution can redeem his honor and Dulcinea's beauty. Therefore, he sees his goals as no longer attainable. The subsequent step in Don Quijote's return to sanity is the revelation of the victorious knight as Sansón Carrasco, who believes

"'his health is in his rest and in that he be on his lands and in his house, I planned a scheme to make him be there'" (ibid., II.LXV.1049). Carrasco's plan to defeat Don Quijote to return him to sanity succeeds, in that the knight and squire discuss plans for the future that do not involve knight errantry. Don Quijote's delusions about chivalric romance cannot persist, and so the mad knight returns home and falls into a depression, which confines him to his bed for six days. After this, Don Quijote attests to his return to his former self:

> —Give me gifts for good news, good sirs, that I am no longer don Quijote de la Mancha, rather Alonso Quijano, whom my familiars gave the epithet 'the good.' I am enemy to Amadís of Gaul and the whole infinite bunch of his lineage; already they are odious to me, all the profane stories of knight errantry; I know my folly and the danger in which they put me having read them; for the mercy of God chastising my right mind, I abhor them. (ibid., II.LXXIV.1100-1)

From this passage, the then–and–now dichotomy clarifies that the madness was connected to the notion of a contemporary chivalric romance as a means to demonstrate its futility in the contemporary world. Those characters whom Don Quijote emulated Alonso Quijano now abhors. The anachronism is also invoked in this return of Quijano's senses when he mentions how circumstances have changed: "'in the nests of yesteryear are no birds of this year. I was crazy and now I am sane'" (ibid., II.LXXIV.1103). The primary acknowledgement of change pertains to his madness, and so the metonymic connection between madness and chivalry has been lost. However, the political critiques of the contemporary state of the Spanish empire persist, as they have been disguised in the multivalent narrative device of madness.

## 3. Discussion

Since madness masks the paradigm shift, *Don Quijote* avoids censorship, though it satirizes the association of chivalric romance with imperial practices. Cervantes's implementation of madness to publish social critique without censorship indicates the liminality of madness, and disability in general, as an entity in society: one with which and through which anachronistic and undesirable values could be associated and critiqued in order to encourage paradigmatic progress and advancement. This precedent for madness representation reflects literary disability devices: an aesthetic nervousness that gives way to narrative prosthesis, or "a crutch upon which literary narratives lean for their representational power, disruptive potentiality, and analytical insight" (Quayson 2007; Snyder and Mitchell 2014, p. 49). The social critique in *Don Quijote* employs disability aesthetics to advance its anti-imperialist position. Michael Bérubé writes on the legacy of mental disability in literature as one that advances frontiers in thought and ideology, since:

> Many of the narratives in the world of literature. . .manage to make thematic or formal use of intellectual disability without entangling themselves in bewildering disability chronotopes or metafictional Möbius strips. And yet when such narratives *do* entangle themselves in bewildering disability chronotopes, they open onto stunning spatiotemporal vistas that exceed ordinary human comprehension (and can be accessed only by extraordinary human comprehension). (Bérubé 2016, pp. 158–59, original italics)

In other words, madness invokes a paradox in literary representation: it is associated with sociocultural liminalities such as passé social structures and therefore is rejected from the center, and yet it is inextricable from the center of sociocultural progress as a device in literature whose aim is to improve human thinking. Disability has been at the center of social criticism as a way of perceiving those who have less power than others in order to redistribute influence and restructure society to accommodate those with less power. However, these gestures have not empowered the disabled, rather, further suppressed them from the Early Modern European image of the human.

The use of disability metonymy in Early Modern literature curiously manages to broach anxieties around race, gender roles, and embodiment, all while evading censorship.

These texts engage Islamic alterity during the Hapsburg empire to invoke the colonial and imperial practices that have taken hold of European societies, which allows the further association of chivalry with imperial practices throughout their satires. The resistant or illusory nature of the ideal woman for these mad knights calls into question the societal correspondence of imperial interactions with the world in the Early Modern period. The trouble with the paradox of chivalric madness is that in satirizing the revival of chivalry to promote the imperial framework bolstered by empiricist and rationalist epistemologies under development by Cervantes's time, madness representations reinforce the positioning of disabled and mad embodiments as a literary device central to satire, an uncensored critique of contemporary epistemologies. Once established, these literary devices risk shaping later European understandings of deviance, both literary and real, from the empiricist ideal embodiment with a colonization of the body in mind.

**Funding:** This research received no external funding.

**Data Availability Statement:** The original contributions presented in the study are included in the article, further inquiries can be directed to the corresponding author/s.

**Conflicts of Interest:** The author declares no conflict of interests.

## Notes

1   These legal protections were important, because another writer did publish his own version of *la Segunda Parte*, which prompted Cervantes to write has second installment a decade later.
2   "Normate" comes to critical disability studies through Rosemarie Garland Thomson's *Extraordinary Bodies*, where she identifies it as "the veiled subject position of cultural self, the figure outlined by the array of deviant others...the constructed identity of those who, by way of the bodily configurations and cultural capital they assume, can step into a position of authority and wield the power it grants them" (Thomson 1997, p. 8).

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
