# Peer review of "The Paradox of Chivalric Madness: Ariosto’s and Cervantes’s Madness Representations’ Impact on Disability Representation"

_humanities, doi:10.3390/h13030087_

Round 1
Reviewer 1 Report
Comments and Suggestions for Authors
There are two basic problems: the confusion of the early modern periods of "Renaissance" and "Baroque" with "Enlightenment," and the lack of any mention of the disability that Cervantes mentions in his Prologue: in the battle of Lepanto he lost the use of his left hand.
Ariosto was writing during the period traditionally known as the Italian Renaissance; Cervantes between Renaissance and Baroque. The moral problems of imperialism in Spain were questioned from the conquest of Mexico (Las Casas), with arbitristas writing about concern for the effect on the Spanish economy in the mid-1600s.
Bearden's use of sprezzatura needs explanation when it is first mentioned.
Line 182—the opening lines depend on readers’ familiarity with Virgil’s Aeneid: Arma virumque cano (I sing of arms and the man)
Lines 210-212—incomplete sentence, but also needs an explanation: why “love is not involved”
Lines 233-239—why is Italian prose quoted without translation while Italian poetry is quoted in English translation, without the original, and Cervantes’s Spanish is quoted in the original?
Line 292—Angelica is pagan because she is from "Cathay," a place identified in literary works as located in various parts of in Asia.
Line 429—“giving way to imperialist thought”? Spain had already had a century of conquest and colonization in the Americas, and arguments about the morality of empire (e.g. Las Casas). In Cervantes’s time, there was concern about the abuses of imperialism and its deleterious effect on the Spanish economy.
Comments on the Quality of English Language
Line 83—delete “in”
Line 126—“moreso”?
Line 209—constructions of obedience
Line 562--incomplete sentence ending with "since."
Author Response
- Are the research design, questions, hypotheses and methods clearly stated? must be improved
- Thank you for this note. I have expanded the research questions section in the introduction and worked toward clarifying the theoretical framework employed throughout the study. If there are other issues with the questions and methods for this reviewer, I would appreciate further elaboration on how to adjust them for publication.
- There are two basic problems: the confusion of the early modern periods of "Renaissance" and "Baroque" with "Enlightenment," and the lack of any mention of the disability that Cervantes mentions in his Prologue: in the battle of Lepanto he lost the use of his left hand. Ariosto was writing during the period traditionally known as the Italian Renaissance; Cervantes between Renaissance and Baroque. The moral problems of imperialism in Spain were questioned from the conquest of Mexico (Las Casas), with arbitristas writing about concern for the effect on the Spanish economy in the mid-1600s.
- Thank you for this note. I have consolidated and clarified the temporal as well as philosophical references to be more precise as to what I’m referring in the epistemological timeline. Also, I had not yet included work that saw Cervantes as a literary arbitrista, and so I added it to enter more into this dialogue. However, I would like to hear more on the significance of Cervantes’s disclosed disability, as that biographical criticism does not have much scholarship that would make it relevant to his portrayal of madness. If the reviewer could elaborate on how this should factor in, I would greatly appreciate that so I can adjust the study accordingly.
- Bearden's use of sprezzatura needs explanation when it is first mentioned.
- I have fleshed out the definition given of this concept and hope it is clearer. If the reviewer still has questions or concerns about the definition, I would appreciate learning of it.
- Line 182—the opening lines depend on readers’ familiarity with Virgil’s Aeneid: Arma virumque cano (I sing of arms and the man)
- An excellent intertextual point to consider, especially given I am relying on Ariosto’s intertextual world to point at the discourse he’s entering. I have included this idea and worked on elaborating the intertextuality as part of its significance.
- Lines 210-212—incomplete sentence, but also needs an explanation: why “love is not involved”
- I have reorganized that sentence to clarify the position I am taking.
- Lines 233-239—why is Italian prose quoted without translation while Italian poetry is quoted in English translation, without the original, and Cervantes’s Spanish is quoted in the original?
- I have changed all non-English language quotations into English and added notification of places where I had to translate.
- Line 292—Angelica is pagan because she is from "Cathay," a place identified in literary works as located in various parts of in Asia.
- This is a wonderful point to tease out in the study here, since it is unclear from the text whether she is from Cathay or from India. I have worked on emphasizing the posturing of Angelica and its similarity to Africa Devicta at this time.
- Line 429—“giving way to imperialist thought”? Spain had already had a century of conquest and colonization in the Americas, and arguments about the morality of empire (e.g. Las Casas). In Cervantes’s time, there was concern about the abuses of imperialism and its deleterious effect on the Spanish economy.
- Thank you for raising this point so I could come back to it. I must have gotten mixed up in the revisions before submission of the draft. I have edited this passage to better reflect the controversies surrounding imperialism contemporary with Cervantes.
Reviewer 2 Report
Comments and Suggestions for Authors
The Enlightenment comes after Ariosto and Cervantes.
I personally am not convinced about this article. The topic is relevant, even fashionable, and the conclusion about the legacy of these 2 works for literary representations of disability is interesting and probably not much studied up to now. However, an article about anachronism should not be anachronistic. The author constantly refers to Ariosto, Cervantes, even Erasmus as figures of the Enlightenment or examples of the Enlightenment paradigm. As a minimum condition of publication, every reference to the Enlightenment should be suppressed. Moreover, there is a complete misuse of the concept of sprezzatura from Castiglione, where it connotes a sort of non-challance. Also the author doesn’t seem to know the meaning of antiphrasis, or saying the opposite of what you mean. Another annoying feature is the gratuitous appeal to absent philosophies, “empiricist and Cartesian epistemologies.” Finally, we hear a lot about DQ’s anti-imperialist position, but it is never demonstrated (though it could be).
Comments on the Quality of English Languageone more proofreading wouldn't hurt
Author Response
- Are the arguments and discussion of findings coherent, balanced and compelling? Must be improved
- Thank you for this prompt to focus my argument further. I think by focusing more precisely I have accomplished a stronger presentation of findings in the study. If there are further problems with the presentation of findings, I would very much like to know more about what pitfalls this reviewer finds in the argumentation.
- However, an article about anachronism should not be anachronistic. The author constantly refers to Ariosto, Cervantes, even Erasmus as figures of the Enlightenment or examples of the Enlightenment paradigm. As a minimum condition of publication, every reference to the Enlightenment should be suppressed.
- I have consolidated the philosophical entities to be more concerned with the empiricism and rationalism that was emergent during the Baroque than the Enlightenment. Any further appearances of the Enlightenment in the draft are unintentional.
- Moreover, there is a complete misuse of the concept of sprezzatura from Castiglione, where it connotes a sort of non-challance.
- The use of sprezzatura is coming from Elizabeth Bearden’s book in which it features prominently. By not referring back to Castiglione, I may have been in error, but for the purposes of this study, Bearden’s definition and work on sprezzatura was what was most prudent.
- Also the author doesn’t seem to know the meaning of antiphrasis, or saying the opposite of what you mean.
- I have substituted antiphrasis with hyperbole which should find the meaning more directly.
- Another annoying feature is the gratuitous appeal to absent philosophies, “empiricist and Cartesian epistemologies.”
- By cutting down on terms, I have consolidated and hopefully defined the philosophies I do use to be less absent and more pertinent to the work.
- Finally, we hear a lot about DQ’s anti-imperialist position, but it is never demonstrated (though it could be).
- I have added a paragraph that begins at line 534 to address the anti-imperialism and its relationship to madness. I hope it helps to tie some of the findings together.
Round 2
Reviewer 1 Report
Comments and Suggestions for Authors
The revisions and addition of Nemser's discussion are well integrated into the paper.
Reviewer 2 Report
Comments and Suggestions for Authors
I think it passes the test this time.